# Data-driven optimization of diet formulation to enhance survival and growth in Japanese Eel (*Anguilla japonica*) larvae

**Kazuharu Nomura**[ID][1]\*, **Tadao Jinbo**[2], **Hirofumi Furuita**[1], **Masato Higuchi**[2], **Taiga Asakura**[3], **Hiroshi Suzuki**[2], **Takashi Yatabe**[2], **Miyuki Mekuchi**[3], **Takeshi Hano**[4], **Takashi Ishikawa**[1], **Youhei Fukui**[1], **Nobuto Kaneko**[ID][1]

**1** Fisheries Technology Institute Nansei Field Station, Japan Fisheries Research and Education Agency, Nakatsuhamaura, Minamiise, Watarai, Mie, Japan, **2** Fisheries Technology Institute Shibushi Field Station, Japan Fisheries Research and Education Agency, Natsui, Shibushi, Kagoshima, Japan, **3** Fisheries Resources Institute Yokohama Field Station, Japan Fisheries Research and Education Agency, Fukuura, Kanazawa, Yokohama, Kanagawa, Japan, **4** Fisheries Technology Institute Hatsukaichi Field Station, Japan Fisheries Research and Education Agency, Maruishi, Hatsukaichi, Hiroshima, Japan

\* nomura_kazuharu57@fra.go.jp

## Abstract

Developing an optimal larval diet for Japanese eel (*Anguilla japonica*) remains a major bottleneck in artificial seedling production due to the limited efficacy of conventional heuristic formulation methods. We implemented a data-driven, human-in-the-loop optimization framework based on Bayesian optimization (BO) and Gaussian process regression (GPR) to systematically refine shark egg yolk-free diets. Across eight sequential feeding trials, the ingredient ratios of seven key components were iteratively adjusted, guided by empirical expert insight. Survival rate and total length (TL) were measured from 6 to 100 days post-hatch (dph). Multi-objective optimization targeted simultaneous improvement in survival and growth during the early (6–40 dph) and late (41–80 dph) periods. The optimized diet formulations achieved survival rates up to 74.9% versus 30.7% in controls and mean TL of 40.1 mm versus 35.2 mm ($p < 0.05$) at 100 dph. These results were consistently reproduced in independent validation trials. Mechanistic modeling via Random Forest and SHapley Additive exPlanations (SHAP) revealed that stage-specific balancing of yeast extract-derived nutrients and other macronutrient components were critical drivers of improved survival and growth. To our knowledge, this is the first reported application of BO to larval feed optimization in Japanese eel. Our framework integrates probabilistic modeling with expert input to reduce experimental burden and accelerate development of superior larval diet formulations. This approach offers a practical, broadly applicable template for feed innovation across aquaculture species.

**Data availability statement:** All relevant data are within the paper and its Supporting Information files.

**Funding:** This work was supported by a grant-in-aid entitled "Demonstration Project of a Mass-Production System for the Commercialization of Eel Seedlings," funded by the Fisheries Agency of the Government of Japan. The funders had no role in study design, data collection and analysis, decision to publish, or preparation of the manuscript.

**Competing interests:** The authors have declared that no competing interests exist.

## Introduction

The Japanese eel (*Anguilla japonica*) is a commercially valuable aquaculture species in East Asia. Current aquaculture production remains entirely dependent on wild-caught glass eels for seed stock. To mitigate this dependence, intensive research since the 1960s has aimed to establish a closed-cycle production system, achieving milestones such as hormone-induced maturation [1] and successful larval rearing [2,3]. However, large-scale artificial seedling production is constrained by high mortality during the extended larval phase, typically spanning five to 15 months [4]. Among the various factors influencing larval survival, the absence of an optimized larval diet remains one of the most significant bottlenecks.

The initial artificial propagation of *A. japonica* larvae relied on shark egg yolk-based diets, which enabled the first successful metamorphosis of *A. japonica* into glass eels [2]. However, reliance on shark egg yolk raises sustainability concerns and practical limitations related to supply and cost. Consequently, alternative diets that achieve growth and survival rates comparable to, or surpassing, those of conventional shark egg yolk-based diets have been developed [5,6]. While these advances represent important progress, further improvements are expected through precise optimization of ingredient ratios. The main challenge lies in efficiently exploring high-dimensional formulation spaces to identify truly optimal dietary compositions, while minimizing experimental resources.

Effective optimization of larval diets must account for multiple factors. Diets must fulfill the specific nutritional requirements of eel larvae, including essential amino acids, fatty acids, vitamins, and minerals, while also being suited to their unique digestive physiology. Eel larvae lack a functional stomach [7] and rely primarily on pancreatic enzymes for digestion [8]. Physical properties of the diet, such as particle size and viscosity, are equally critical, as overly sticky or dense diets can disrupt normal feeding behavior [9]. Due to the complexity of these interacting factors, conventional feed development based on trial and error or simple screening is often inefficient and resource-intensive.

To address these challenges, adaptive experimental design techniques offer a promising solution for efficiently identifying optimal feed compositions. Earlier studies in aquaculture nutrition introduced mixture-based experimental designs to systematically explore ingredient interactions and improve formulation efficiency [10]. Building on these foundations, recent advances in computational modeling have led to the emergence of data-driven optimization approaches, most notably Bayesian optimization (BO), which have attracted increasing attention in animal diet design [11]. BO utilizes probabilistic surrogate models, such as Gaussian processes, to predict the relationship between feed composition and performance metrics (e.g., growth rate, feed efficiency, and survival) while explicitly quantifying uncertainty [12]. This approach enables efficient exploration of complex feed formulation spaces and can substantially reduce the number of experimental trials required to identify nutritionally optimal and cost-effective diets. A key advantage of BO is its ability to iteratively refine predictions based on accumulated data, enabling the identification of

high-performance formulations within a limited number of trials. Although BO-driven feed optimization has shown benefits in related domains, such as aquaculture [13], poultry [14], and swine [15], to date, no published reports exist of BO being applied to larval feed optimization, including for *A. japonica*.

In this study, we aimed to optimize the larval feed composition of Japanese eels using an adaptive experimental design based on BO, targeting improvements in both growth and survival rates up to 100 days post-hatch (dph). Through a series of feeding trials with varying ingredient ratios, BO was used to iteratively refine the diet formulations. In addition, we employed a Random Forest (RF) regression model with SHapley Additive exPlanations (SHAP) analysis to quantitatively evaluate the contribution of each proximate nutrient component and to estimate the optimal ranges for key dietary factors associated with improved larval survival and growth. This data-driven approach aims to establish a more systematic and efficient framework for larval feed development, thereby contributing to the advancement and sustainability of *A. japonica* aquaculture.

## Materials and methods

### Ethics statement

All experimental procedures were conducted in accordance with the Guidelines for Animal Experimentation at the Fisheries Technology Institute, Japan Fisheries Research and Education Agency. Experimental protocols were approved by the Institutional Animal Care and Use Committee of the Fisheries Technology Institute (approval codes: 21008, 22016, and 23015). Humane endpoints were established to minimize potential suffering of larvae. Individuals that exhibited cessation or markedly sluggish movement for more than 30 s, complete loss of body transparency, whitening of the head and notochordal nerve, or severe spinal curvature were considered moribund and immediately removed from the tank. These larvae were euthanized by an overdose of 2-phenoxyethanol (> 1,000 ppm) and disposed of according to institutional guidelines. Larval conditions were monitored at least three times daily throughout the experiments. No unexpected deaths occurred beyond those anticipated for normal survival assessments.

### Experimental fishes and rearing conditions

Adult female and male Japanese eels were hormonally induced to mature at the Shibushi Field Station of the Fisheries Technology Institute, Japan Fisheries Research and Education Agency, Kagoshima, Japan. Wild glass eels were purchased from commercial suppliers and feminized by feeding a commercial diet containing estradiol-17β (E2; 10 mg/kg feed) for 6 months [16], followed by a standard commercial diet until the onset of artificial maturation. Males were either purchased from commercial eel farms or obtained by rearing artificially produced juveniles. Mature gametes were obtained as described by [17]. For each trial, eggs from a single female were fertilized in vitro using a sperm mixture from 5–8 males. Fertilized eggs were maintained in cylindrical polyester nets in tanks with flowing seawater at 25±0.5 °C until hatching. Hatched larvae were transferred to a 100 L round-bottom tank and kept unfed until 5 dph, after which 250 larvae were distributed into each of up to twelve 10 L acrylic bowl tanks, depending on the experimental design. All tanks were maintained at 23±0.5 °C. Other rearing conditions were as described by Jinbo et al. (2025). Feeding commenced at 6 dph and continued until 80 or 100 dph, depending on the trial phase.

### Diet formulations and preparation

All experimental diets were based on a previously developed shark egg yolk-free formulation comprising nine ingredients: egg yolk powder, skimmed milk powder, yeast extract, soybean peptide, casein sodium, fish protein hydrolysate (CPSP), fish oil, vitamin mix, and taurine. The detailed composition of each diet is presented in Table 1. Diets were prepared by mixing the powdered ingredients with 1.7–3.9 volumes of distilled water to adjust viscosity, homogenizing with a blender (A-1100SL, Acasas, Hyogo, Japan), and then stirring and deaeration using a planetary centrifugal mixer (ARE-310, THINKY CORPORATION, Tokyo, Japan) to yield a homogeneous slurry-type diet. The diets were stored at −20 °C until use. Two standard reference diets, FSD and FSY (D1 and D6 in Jinbo et al. [6], with slight modification), were included in all trials.

**Table 1. Formulation of the experimental diets (%).**

| Diet_ID | Egg yolk powder[1] | Skimmed milk powder[2] | Yeast extract[3] | Soy peptide[4] | Casein Na[5] | CPSP[6] | Fish oil[7] | Vitamin mix[8] | Tau-rine[9] | Total (%) |
|---|---|---|---|---|---|---|---|---|---|---|
| No. 1 | 16.1 | 10.8 | 0.0 | 10.8 | 10.8 | 43.0 | 4.3 | 2.2 | 2.2 | 100.0 |
| No. 2 | 10.8 | 10.8 | 8.1 | 8.1 | 21.5 | 32.3 | 4.3 | 2.2 | 2.2 | 100.0 |
| No. 3 | 2.7 | 0.0 | 26.9 | 0.0 | 32.3 | 33.9 | 0.0 | 2.2 | 2.2 | 100.0 |
| No. 4 | 9.7 | 16.1 | 0.0 | 14.0 | 32.3 | 21.5 | 2.2 | 2.2 | 2.2 | 100.0 |
| No. 5 | 16.1 | 16.1 | 26.9 | 0.0 | 10.8 | 25.8 | 0.0 | 2.2 | 2.2 | 100.0 |
| No. 6 | 2.7 | 0.0 | 26.9 | 16.1 | 24.2 | 21.5 | 4.3 | 2.2 | 2.2 | 100.0 |
| No. 7 | 5.4 | 0.0 | 16.1 | 16.1 | 31.2 | 25.8 | 1.1 | 2.2 | 2.2 | 100.0 |
| No. 8 | 2.7 | 16.1 | 12.4 | 5.4 | 15.6 | 43.0 | 0.5 | 2.2 | 2.2 | 100.0 |
| No. 13 | 16.1 | 16.1 | 0.0 | 16.1 | 41.9 | 5.4 | 0.0 | 2.2 | 2.2 | 100.0 |
| No. 14 | 16.1 | 0.0 | 0.0 | 0.0 | 43.0 | 36.6 | 0.0 | 2.2 | 2.2 | 100.0 |
| No. 15 | 0.0 | 0.0 | 26.9 | 16.1 | 10.8 | 41.9 | 0.0 | 2.2 | 2.2 | 100.0 |
| No. 16 | 0.0 | 16.1 | 0.0 | 16.1 | 43.0 | 14.0 | 6.5 | 2.2 | 2.2 | 100.0 |
| No. 17 | 0.0 | 16.1 | 26.9 | 0.0 | 40.9 | 5.4 | 6.5 | 2.2 | 2.2 | 100.0 |
| No. 18 | 0.0 | 16.1 | 19.4 | 0.0 | 10.8 | 43.0 | 6.5 | 2.2 | 2.2 | 100.0 |
| No. 19 | 0.0 | 9.7 | 0.0 | 0.0 | 43.0 | 43.0 | 0.0 | 2.2 | 2.2 | 100.0 |
| No. 20 | 16.1 | 0.0 | 26.9 | 0.0 | 40.9 | 5.4 | 6.5 | 2.2 | 2.2 | 100.0 |
| No. 21 | 16.1 | 16.1 | 26.9 | 16.1 | 10.8 | 9.7 | 0.0 | 2.2 | 2.2 | 100.0 |
| No. 22 | 16.1 | 0.0 | 3.2 | 16.1 | 10.8 | 43.0 | 6.5 | 2.2 | 2.2 | 100.0 |
| No. 23 | 16.6 | 8.4 | 2.1 | 10.1 | 29.7 | 28.8 | 0.0 | 2.1 | 2.1 | 100.0 |
| No. 24 | 25.7 | 21.0 | 3.2 | 1.0 | 0.6 | 43.7 | 0.4 | 2.2 | 2.2 | 100.0 |
| No. 25 | 26.5 | 1.1 | 0.1 | 13.1 | 43.7 | 9.9 | 1.3 | 2.2 | 2.2 | 100.0 |
| No. 26 | 24.0 | 1.7 | 23.6 | 0.5 | 15.8 | 29.7 | 0.5 | 2.1 | 2.1 | 100.0 |
| No. 27 | 26.7 | 1.6 | 9.9 | 5.4 | 5.8 | 46.2 | 0.2 | 2.2 | 2.2 | 100.0 |
| No. 28 | 2.2 | 0.4 | 21.4 | 3.5 | 21.4 | 46.7 | 0.1 | 2.1 | 2.1 | 100.0 |
| No. 29 | 16.2 | 0.0 | 0.0 | 0.0 | 43.0 | 36.6 | 0.0 | 2.2 | 2.2 | 100.0 |
| No. 30 | 8.6 | 0.0 | 26.9 | 0.0 | 10.7 | 43.0 | 6.5 | 2.2 | 2.2 | 100.0 |
| No. 31 | 19.4 | 19.4 | 15.1 | 0.0 | 38.7 | 0.0 | 3.2 | 2.2 | 2.2 | 100.0 |
| No. 33 | 19.4 | 20.4 | 15.1 | 0.0 | 40.9 | 0.0 | 0.0 | 2.2 | 2.2 | 100.0 |
| No. 34 | 15.8 | 24.6 | 1.7 | 13.0 | 39.5 | 0.5 | 0.5 | 2.2 | 2.2 | 100.0 |
| No. 35 | 11.8 | 3.8 | 10.3 | 13.0 | 37.5 | 17.5 | 1.7 | 2.2 | 2.2 | 100.0 |
| No. 36 | 10.8 | 18.9 | 18.7 | 0.8 | 37.7 | 2.3 | 6.6 | 2.2 | 2.2 | 100.0 |
| No. 37 | 22.3 | 5.5 | 3.9 | 0.1 | 45.2 | 13.1 | 5.7 | 2.2 | 2.2 | 100.0 |
| No. 38 | 19.9 | 8.5 | 7.7 | 0.9 | 46.9 | 7.8 | 4.0 | 2.2 | 2.2 | 100.0 |
| No. 39 | 17.0 | 14.2 | 8.4 | 1.0 | 22.7 | 25.8 | 6.7 | 2.2 | 2.2 | 100.0 |
| No. 40 | 10.3 | 5.7 | 9.0 | 7.3 | 33.5 | 27.5 | 2.3 | 2.2 | 2.2 | 100.0 |
| No. 41 | 18.6 | 7.0 | 7.8 | 3.1 | 40.0 | 15.9 | 3.3 | 2.2 | 2.2 | 100.0 |
| No. 42 | 25.3 | 5.5 | 17.8 | 0.1 | 27.3 | 18.9 | 0.8 | 2.2 | 2.2 | 100.0 |
| No. 43 | 25.5 | 2.8 | 25.5 | 0.5 | 26.1 | 10.3 | 4.9 | 2.2 | 2.2 | 100.0 |
| No. 44 | 22.1 | 20.2 | 13.5 | 0.2 | 33.4 | 4.8 | 1.4 | 2.2 | 2.2 | 100.0 |
| FSD (reference 1) | 16.0 | 16.0 | 0.0 | 5.3 | 21.3 | 31.9 | 5.3 | 2.1 | 2.1 | 100.0 |
| FSY (reference 2) | 10.8 | 10.8 | 16.1 | 0.0 | 21.5 | 32.3 | 4.3 | 2.2 | 2.2 | 100.0 |

[1]Dried egg yolk No. 1 (Kewpie Egg, Tokyo, Japan).

[2]Hokkaido skim milk (Yotsuba Milk Products, Sapporo, Japan).

[3]Mixture of AROMILD™ and Yeast Extract NT (KOHJIN Life Sciences Co., Ltd, Tokyo, Japan) at a ratio of 1:1.

[4]Hipolypepton N (Shiotani M.S., Hyogo, Japan).

*(Continued)*

**Table 1.** (Continued)

[5]Sodium caseinate LW (Arla, Viby J, Denmark).

[6]CPSP Special G (Sopropeche, Wimille, France).

[7]Highcalor E (Kanematsu Shintoa Foods, Tokyo, Japan).

[8]Japan Nutrition, Tokyo, Japan, Vitamin mixture composition (g or IU/kg): Vitamin A, 220,000 IU; Vitamin D3, 90,000 IU; dl-α-tocopherol acetate, 6.000 g; Menadione sodium bisulfite, 2.302 g; Thiamine nitrate, 9.255 g; Riboflavin, 6.000 g; Pyridoxine hydrochloride, 7.296 g; Nicotinamide, 22.317 g; Calcium pantothenate, 16.311 g; Folic acid, 0.450 g; Cyanocobalamin, 0.015 g; Biotin, 0.150 g; Inositol, 60.000 g; Calcium ascorbate, 108.962 g.

[9]Nacalai Tesque (Kyoto, Japan).

## Experimental design and adaptive optimization process

Eight consecutive feeding trials (Tr_1–8; Table 2) were conducted across four phases, combining a D-optimal experimental design with BO to iteratively improve the shark egg yolk-free larval diet.

**Phase I – Baseline data collection and D-optimal design (Tr_1–2).** To efficiently select diet formulations for initial data collection, a two-step D-optimal experimental design was generated using JMP version 16 (SAS Institute Inc., Cary, NC, USA). Seven major ingredients (egg yolk powder, skim milk powder, yeast extract, soybean peptide, casein sodium, CPSP, and fish oil) were set as variables, while vitamin mix and taurine proportions were held constant. Based on the design outputs, 18 formulations (Nos. 1-8 and 13-22) were finalized through minor manual adjustments to ensure physicochemical feasibility (e.g., slurry consistency suitable for larval ingestion and adequate binding stability in water) and biological safety (i.e., avoidance of extreme nutrient imbalances), while remaining within the predefined boundaries summarized in S1 Table. These diets were evaluated alongside two reference diets (FSD and FSY) in single-tank trials. In Tr_1, eight preliminary diets (Nos. 1-8) were tested using initial feasible ingredient ranges. The outcomes of Tr_1 were subsequently used to refine ingredient constraints, and ten additional formulations (Nos. 13-22; Nos. 9-12 were excluded from the present dataset) were generated and tested in Tr_2.

**Phase II – BO exploration (Tr_3–4).** Based on the Tr_1 and Tr_2 results, which suggested that the optimal diet formulation differed by growth stage, the dataset used for BO from Tr_3 onward was divided into two periods: 6–40 dph and 41–80 dph. For each period, two objective variables, survival rate and mean TL, were modeled independently. The explanatory variables were the proportions of the seven ingredients normalized to a total sum of one. For each trial, the data for each diet (i.e., tank) were normalized to a standardized z-score, calculated as the deviation from the mean of the two reference diets divided by the standard deviation (SD) within each trial. Gaussian process regression (GPR) models were constructed independently for each objective variable, excluding explanatory variables with zero variance. Prior to model fitting and prediction, both the explanatory and objective variables were auto-scaled (i.e., centered to zero mean and scaled to unit variance) using statistics calculated from the training data. The optimal kernel function was selected from multiple candidates using 10-fold cross-validation to maximize the predictive $R^2$. Model performance was evaluated using $R^2$, root mean squared error (RMSE), and mean absolute error (MAE) for both the training and cross-validation datasets. In each optimization round, the predictive mean and SD for each candidate formulation were calculated for both survival and TL models. The probability of improvement (PI) was computed using the cumulative distribution function (CDF) of the normal distribution. To ensure balanced improvement in both survival and growth, the sum of the logarithms of the two PI values [i.e., log (PI_survival) + log (PI_TL)] served as the multi-objective acquisition function, with equal weighting assigned to both traits. This approach was adopted to identify candidate formulations that performed well in both objectives. The relaxation parameter for the PI calculation was adjusted according to the experimental phase within the range of $10^{-3}$ to $10^{-6}$; a larger value promoted exploration, whereas a smaller value favored exploitation. For each period, 100 candidate diet formulations were generated based on this combined PI, from which 17 new formulations (Nos. 23-40) were selected using expert judgment and evaluated. After each trial, the newly obtained results were added to

**Table 2. Overview of the eight feeding trials (Tr_1-8) conducted to optimize a larval diet for Japanese eel, *A.japonica*.**

| Trial | Experimental phase/ objective | Diet formulations tested[1] | Formulation origin | Replicates (tanks per diet) | Total no. of tanks | Rearing period/ sampling points (dph)[2] | Primary endpoints[3] |
|---|---|---|---|---|---|---|---|
| Tr_1 | Phase I – collect baseline data | Nos. 1–8 + 2 reference diets (FSD, FSY) | D-optimal design + manual | 1 | 10 | 6-80 (20, 40, 60, 80) | Survival rate, total length |
| Tr_2 | Phase I – collect baseline data | Nos. 13–22 + 2 reference diets (FSD, FSY) | D-optimal design + manual | 1 | 12 | 6-80 (20, 40, 60, 80) | Survival rate, total length |
| Tr_3 | Phase II – Bayesian optimization (BO) exploration | Nos. 23–31 + 2 reference diets (FSD, FSY) | BO-derived | 1 | 11 | 6-80 (20, 40, 60, 80) | Survival rate, total length |
| Tr_4 | Phase II – BO exploration | Nos. 31, 33–40 + 2 reference diets (FSD, FSY) | BO-derived | 1 | 11 | 6-80 (20, 40, 60, 80) | Survival rate, total length |
| Tr_5 | Phase III – BO exploitation & candidate validation | Nos. 41, 41→42, 41→43, 41→44 + 2 reference diets (FSD, FSY) | BO-derived | 2 | 12 | 6-100 (20, 40, 60, 80, 100) | Survival rate, total length |
| Tr_6 | Phase III – BO exploitation & candidate validation (replicate) | Same as Tr_5 | BO-derived | 2 | 12 | 6-100 (20, 40, 60, 80, 100) | Survival rate, total length |
| Tr_7 | Phase IV – Final validation (BO-optimized vs. reference) | Nos.41, 44, 41→44 + 2 reference diets (FSD, FSY) + FSD→FSY | BO-optimized | 2 | 12 | 6-100[4] (20, 40, 60, 80, 100) | Survival rate, total length |
| Tr_8 | Phase IV – Final validation (replicate) | Same as Tr_7 | BO-optimized | 2 | 12 | 6-100[4] (20, 40, 60, 80, 100) | Survival rate, total length |

[1]Two standard reference diets (FSD and FSY; D1 and D6 of Jinbo et al., 2025, slightly modified) were included in every trial. The arrow symbol (→) denotes a feed series in which the diet was switched at 41 dph.

[2]dph = days post-hatch; feeding commenced at 6 dph.

[3]Per sampling, 20–30 larvae/tank were photographed to measure total length (TL); survival rate = survivors/ initial larvae at 6dph. Tank-mean TL and survival rate were the primary endpoints.

[4]To minimise density-related growth bias, larval numbers were equalised to ≤100 individuals per tank at 41 dph in Tr_7–8.

the training dataset, and the BO process was repeated. All computations were performed using Python 3 and standard packages including scikit-learn (source code available in S1 File).

**Phase III – BO exploitation and candidate validation (Tr_5–6).** By exploiting the updated models (favoring exploitation with a smaller relaxation parameter), four leading candidate diet formulation series (Nos. 41, 41→42, 41→43, and 41→44) were selected and tested in duplicate tanks. The rearing period was extended to 100 dph to evaluate consistency and reproducibility. Although a subset of larvae in Tr_5 and Tr_6 were reared beyond 100 dph for supplementary observations (e.g., spinal deformities), these extended data, provided in S4 Table, are not the primary focus of this study.

**Phase IV – Final validation (Tr_7–8).** The top BO-optimized diet formulations (No. 41, 44, and 41→44) were compared head-to-head with the reference diets (FSD, FSY, FSD→FSY) in duplicate tanks up to 100 dph. To minimize density-related growth bias, larval numbers were equalized to ≤ 100 individuals per tank at 41 dph. Details of the diet codes, number of replicates, and sampling schedules for each trial are provided in Table 2.

## Growth and survival assessments

Growth and survival were assessed at 20, 40, 60, 80, and, in later trials, 100 dph. At each time point, 20–30 larvae per tank were randomly selected, anesthetized with 400 ppm 2-phenoxyethanol (FUJIFILM Wako Pure Chemical, Osaka, Japan), photographed using a digital camera (D5600; Nikon, Tokyo, Japan), and returned to their original tank. Total length (TL) was measured from photographs using ImageJ version 1.54 (National Institutes of Health, Bethesda, Maryland, USA). The number of surviving larvae in each tank was determined via direct visual counting. The survival rate was

 

calculated as the percentage of surviving larvae relative to the initial number of stocked larvae per tank (250 individuals). For Tr_7 and Tr_8, the stocking density was adjusted to 41 dph. Thus, after 41 dph, the survival rate was determined by multiplying the cumulative survival rate up to 40 dph by the interval survival rate post-41 dph, using the number of larvae stocked at 41 dph as the new denominator.

### Statistical Analysis

In the final comparative trials (Tr_5–8), differences in TL among diet groups were analyzed by the Tukey-Kramer HSD test, with significance set at $p < 0.05$.

### Analysis of the impact of dietary proximate composition on survival and growth using RF model

To investigate how the proximate composition of the experimental diets influenced larval survival and growth, we analyzed data from all eight feeding trials (Tr_1–8). For each ingredient, the percentages of crude protein, crude lipid, crude ash, crude carbohydrate, and nucleic acids on a dry matter basis were estimated as follows: The moisture content was determined by drying the samples at 110 °C for 10 h. Crude ash was determined by combustion in a muffle oven at 600 °C for 5 h. Crude lipids were extracted using chloroform:methanol (2:1, v/v) according to the method described by [18]. Total nitrogen of the ingredients was determined using the semi-micro Kjeldahl method and crude protein content was calculated as $N \times 6.25$ except for yeast extracts, which contain a high proportion of nitrogen derived from nucleic acids. Protein nitrogen in yeast extracts was calculated by subtracting nucleic acid nitrogen (RNA content $\times 0.146$; [19] from total nitrogen. Nucleic acid content in yeast extracts was determined by high-performance liquid chromatography (HPLC) after extraction with perchloric acid. Crude carbohydrate was calculated as nitrogen-free extract (NFE), using the following equation: NFE = 100 − (moisture + crude protein + crude lipid + crude ash + nucleic acids). Analyses were conducted by the Japan Food Research Laboratories (Tokyo, Japan). Based on these ingredient values (see S2 Table), the proximate composition of each experimental diet was estimated (see S3 Table).

For each period (6–40 dph and 41–80 dph), separate RF regression models were trained for two objective variables: survival rate and mean TL. Prior to model training, the survival rate and mean TL for each diet in each trial were standardized using the mean and standard deviation of the reference diets. The explanatory variables were the estimated percentages (per dry matter) of the four proximate composition components and nucleic acids in each diet. Datasets for 6–40 dph and 41–80 dph were analyzed separately. The hyperparameters of the RF models (e.g., number of trees, maximum depth) were optimized by grid search and five-fold cross-validation to maximize the out-of-bag $R^2$. To interpret the influence of each component, the SHAP values were calculated for each variable. All computations were performed using Python 3, Scikit-learn, and the SHAP package.

## Results

### Baseline data collection and BO exploration (Tr_1–4)

In the initial screening phase (Tr_1–2), 18 experimental diet formulations were evaluated alongside two reference diets (FSD and FSY), and baseline data on larval performance were collected. In the subsequent model-based exploration phase (Tr_3–4), 17 candidate diets were selected using the GPR models based on previous results and were further evaluated. Across these four trials, a total of 35 distinct diet formulations were tested.

Fig 1 summarizes the standardized survival and mean TL (z-scores) for each diet formulation across the four trials, shown separately for three periods: early (6–40 dph), late (41–80 dph), and overall (6–80 dph). Considerable variation was observed among the diet formulations: some achieved much higher survival or growth than the reference diets, whereas others performed less well. Notably, formulation No. 31 showed marked improvements in both survival and growth in Tr_3, serving as a key candidate for further evaluation in subsequent trials. The distribution of z-scores indicated

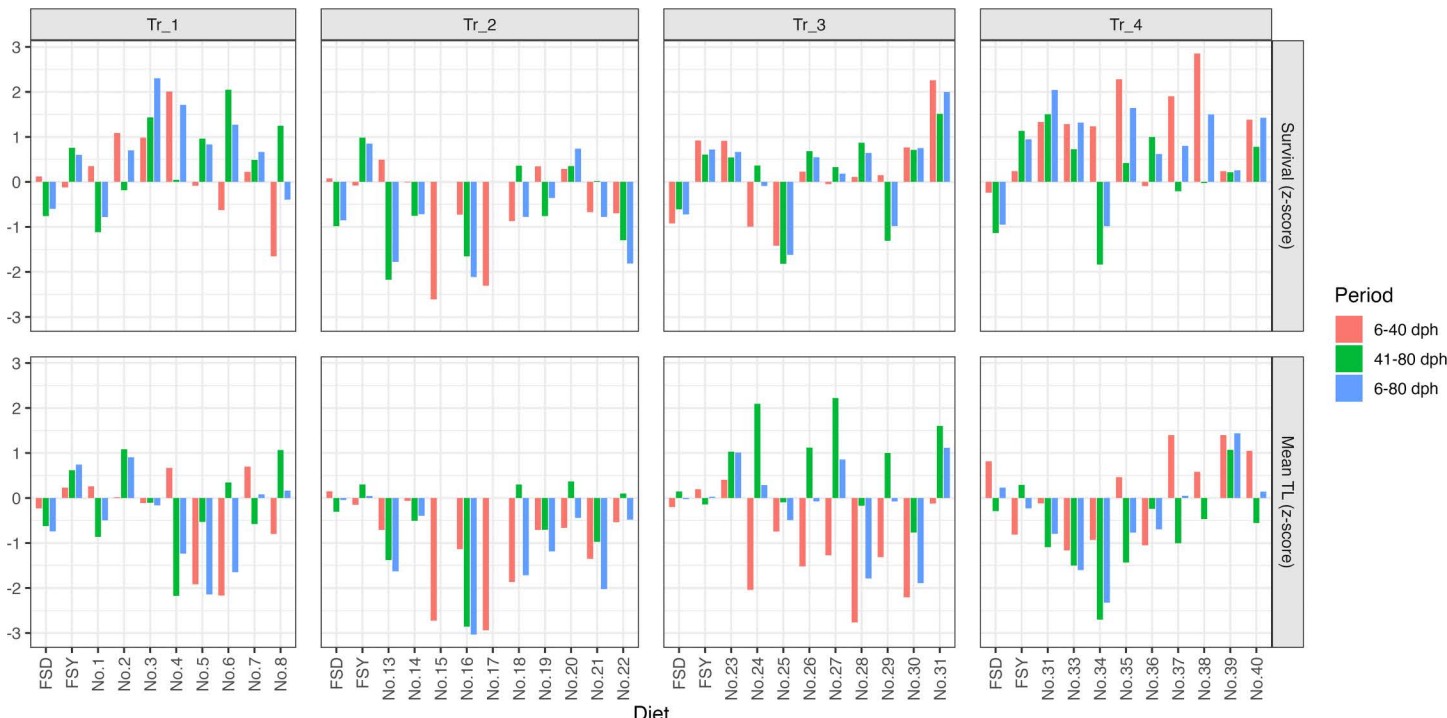

**Fig 1. Standardized survival (upper panels) and mean total length (lower panels) z-scores for each diet formulation across four feeding trials (Tr_1–Tr_4).** Bars represent z-scores for each period (6–40 dph, 41–80 dph, and 6–80 dph), colored as indicated in the legend. Missing data (i.e., diets not tested in certain periods) are not shown.

that improvements in survival and growth did not always coincide across diets and periods, highlighting the importance of evaluating both traits and periods when selecting optimal formulations.

Fig 2 presents the pairwise correlations between standardized survival and mean TL for the three periods, based on data from Tr_1–4. The correlation coefficients between survival and mean TL within each period were weakly positive (6–40 dph: 0.454; 41–80 dph: 0.300; and 6–80 dph: 0.176). In contrast, correlations for the same trait between early and late periods were low or even negative (survival: 0.098; mean TL: −0.172). These results suggest that improvements in survival or growth during the early period do not necessarily lead to better outcomes in the later period. The diversity of responses among diet formulations further highlights the complexity of nutritional requirements at different developmental stages, indicating that the optimal diet composition for maximizing survival and growth likely differs between periods and underscores the need for period-specific optimization strategies.

**Model-based optimization and selection of candidate diets**

Based on the results of the baseline screening, BO was introduced to efficiently search for diet formulations that simultaneously maximized both survival and growth during each period (6–40 dph and 41–80 dph). Standardized z-scores for survival rate and mean TL were used as objective variables, whereas ingredient proportions served as explanatory variables. Four GPR models were constructed for each trait and period. As shown in Table 3, the cross-validated coefficients of determination ($R^2$) for survival and mean TL were 0.261 and 0.472, respectively, during the 6–40 dph period, and 0.748 and 0.320, respectively, during the 41–80 dph period. These results indicate that the model's predictive accuracy varied by period and trait, with the highest accuracy observed for survival in the late period. The RMSE and MAE values,

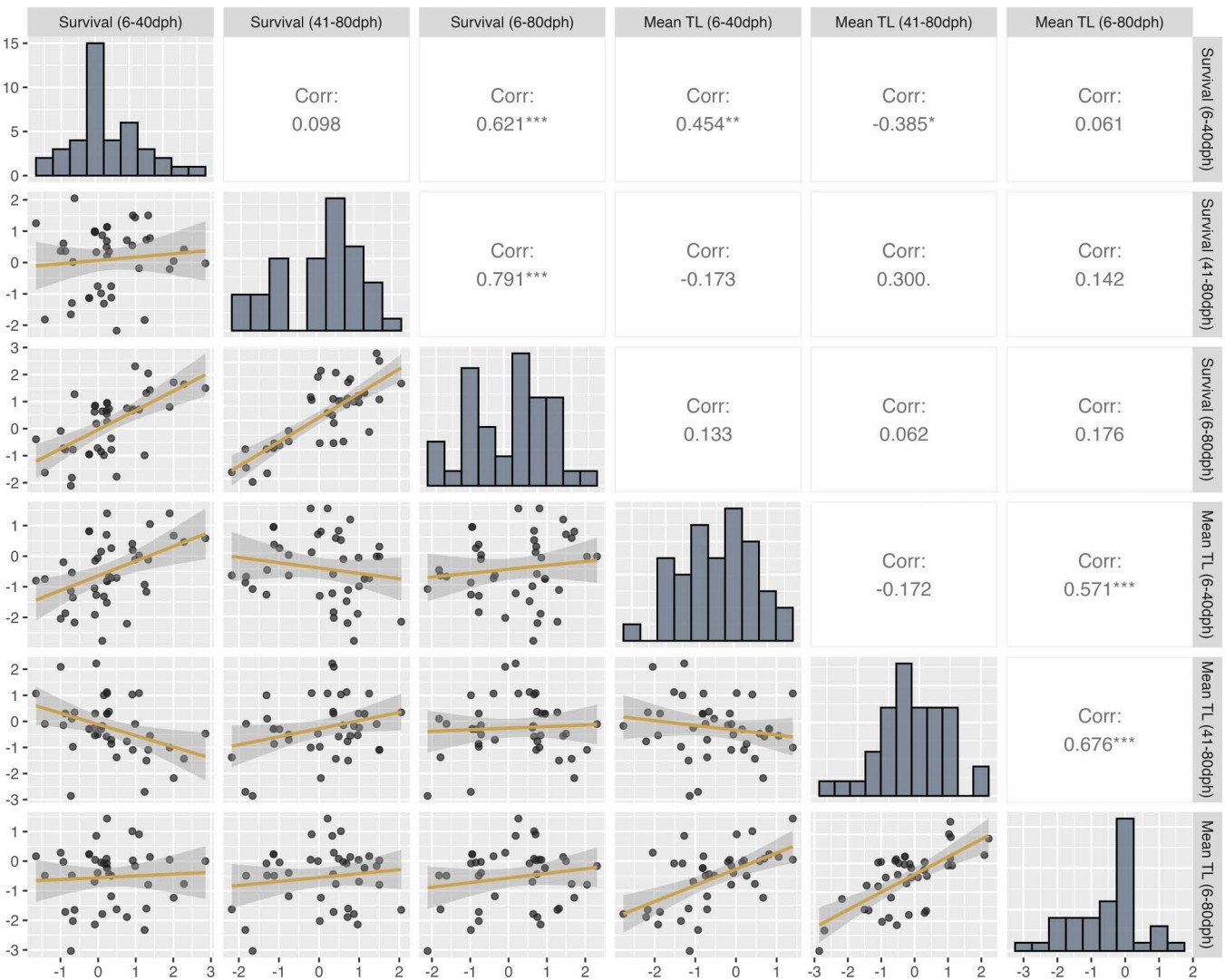

**Fig 2. Pairwise scatterplot matrix of standardized survival and mean total length (z-scores) for Japanese eel larvae across three developmental periods (6–40 dph, 41–80 dph, and 6–80 dph), based on data from feeding trials Tr_1–Tr_4.** Lower panels show scatterplots for each variable pair with colored linear regression lines (solid) and their corresponding 95% confidence intervals (shaded). Diagonal panels display the histogram of each variable. Upper panels indicate Pearson's correlation coefficients (***r***), with significance levels denoted as *$p < 0.05$, **$p < 0.01$, ***$p < 0.001$. Each point represents a single diet formulation.

and optimal kernel functions selected for each model are listed in Table 3. Scatter plots of the predicted versus observed values for all four models are shown in S1 Fig.

Using these models, 100 candidate diet formulations were computationally generated for each period by maximizing the sum of the logarithms of the two PI values for survival and mean TL with equal weighting. A parallel coordinate plot of these candidate formulations is shown in Fig 3. For the 6–40 dph model, most candidate formulations converged to similar ingredient profiles, and the representative average formulation was selected as No. 41. In contrast, for the 41–80 dph model, greater diversity was observed among the candidate formulations. Given resource constraints limiting us to three test candidates (Nos. 42–44), we selected formulations balancing key ingredients (skimmed milk powder, yeast extract, casein sodium, CPSP) to robustly represent the predicted optimal region. Combining these selected diets, four diet series

**Table 3. Result of 10-fold cross-validation by GPR model fitted to data from Tr_1-4.**

| Period | Objective variables | $R^2$ | RMSE | MAE | Selected Kernel Function |
|---|---|---|---|---|---|
| 6-40 dph | Survival rate | 0.261 | 0.944 | 0.722 | $1^2 \times$ RBF (length_scale = 1) + WhiteKernel (noise_level = 1) + $1^2 \times$ DotProduct (sigma$_0$ = 1) |
| | mean TL | 0.472 | 0.619 | 0.609 | $1^2 \times$ Matern (length_scale = 1, ν = 1.5) + WhiteKernel (noise_level = 1) + $1^2 \times$ DotProduct (sigma$_0$ = 1) |
| 41-80 dph | Survival rate | 0.748 | 0.267 | 0.359 | $1^2 \times$ Matern (length_scale = 1, ν = 1.5) + WhiteKernel (noise_level = 1) |
| | mean TL | 0.320 | 0.839 | 0.745 | $1^2 \times$ RBF (length_scale=[1,1,1,1,1,1,1]) + WhiteKernel (noise_level = 1) + $1^2 \times$ DotProduct (sigma$_0$ = 1) |

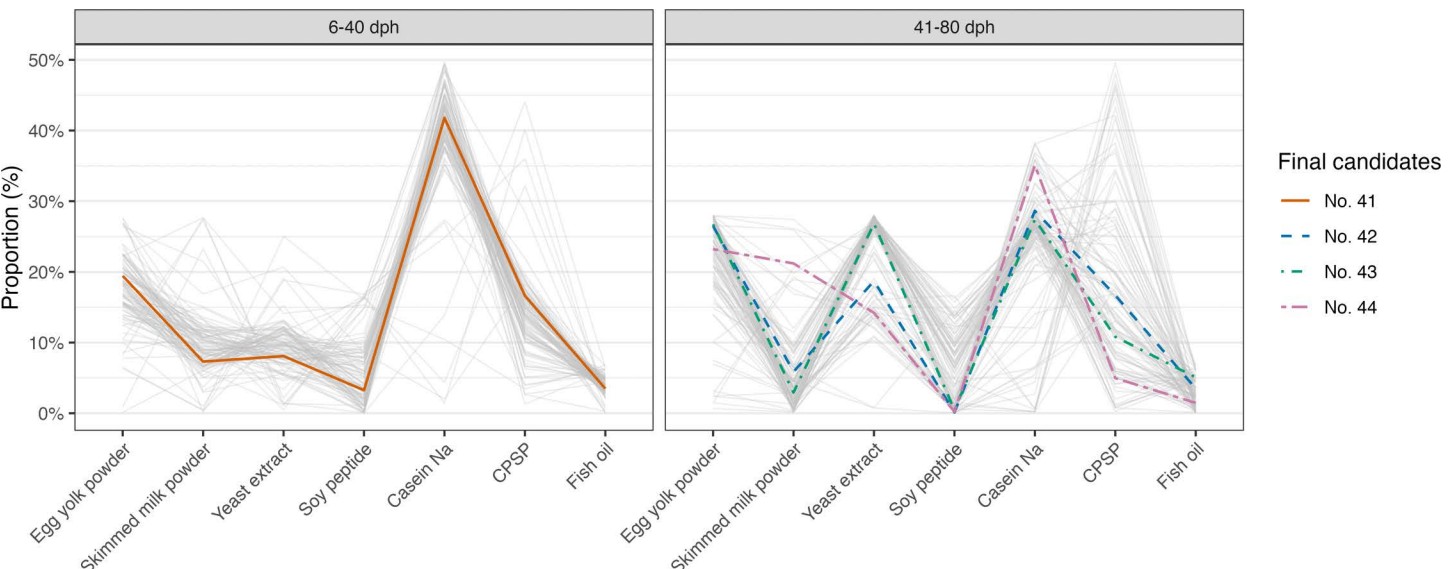

**Fig 3. Parallel coordinate plot of candidate diet formulations generated by BO for each period (6–40 dph and 41–80 dph) using data from Tr_1–Tr_4.** Each line represents the ingredient composition of a single candidate diet, and the vertical axis indicates the percentage of each ingredient (expressed as a proportion out of 100 for all seven ingredients). Gray lines show the distribution of all candidate formulations, while the four diets (Nos. 41, 42, 43, and 44) selected for further validation are highlighted by distinct colors and line types.

(Nos. 41, 41→42, 41→43, and 41→44) were identified as the final candidates and advanced to the validation stage in Tr_5–6.

## Validation and robustness of BO recommended diets (Tr_5–8)

In the validation trials (Tr_5–6), four optimized diet series were tested up to 100 dph with two replicate tanks per treatment. Among the tested diets, the No. 41→44 series yielded the highest mean survival rate at 100 dph in both trials (Tr_5: 66.8%; Tr_6:70.0%), whereas the reference diets showed considerably lower survival rates (FSY: 51.0% in Tr_5 and 57.0% in Tr_6; FSD: 34.0% in Tr_5 and 38.6% in Tr_6; Fig 4A). For mean TL at 100 dph, the No. 41→44 diet achieved high values in both trials (Tr_5: 34.7±1.01 mm; Tr_6: 36.0±0.57 mm), comparable to or slightly below those of the reference diets (FSD: 36.2±0.53 mm in Tr_5, 33.9±0.40 mm in Tr_6; FSY: 35.4±0.64 mm in Tr_5, 33.9±0.70 mm in Tr_6). Tukey-Kramer tests revealed that No. 41→44 was grouped in the top statistical category in each trial, showing no significant difference from the reference diets (Fig 4B, C).

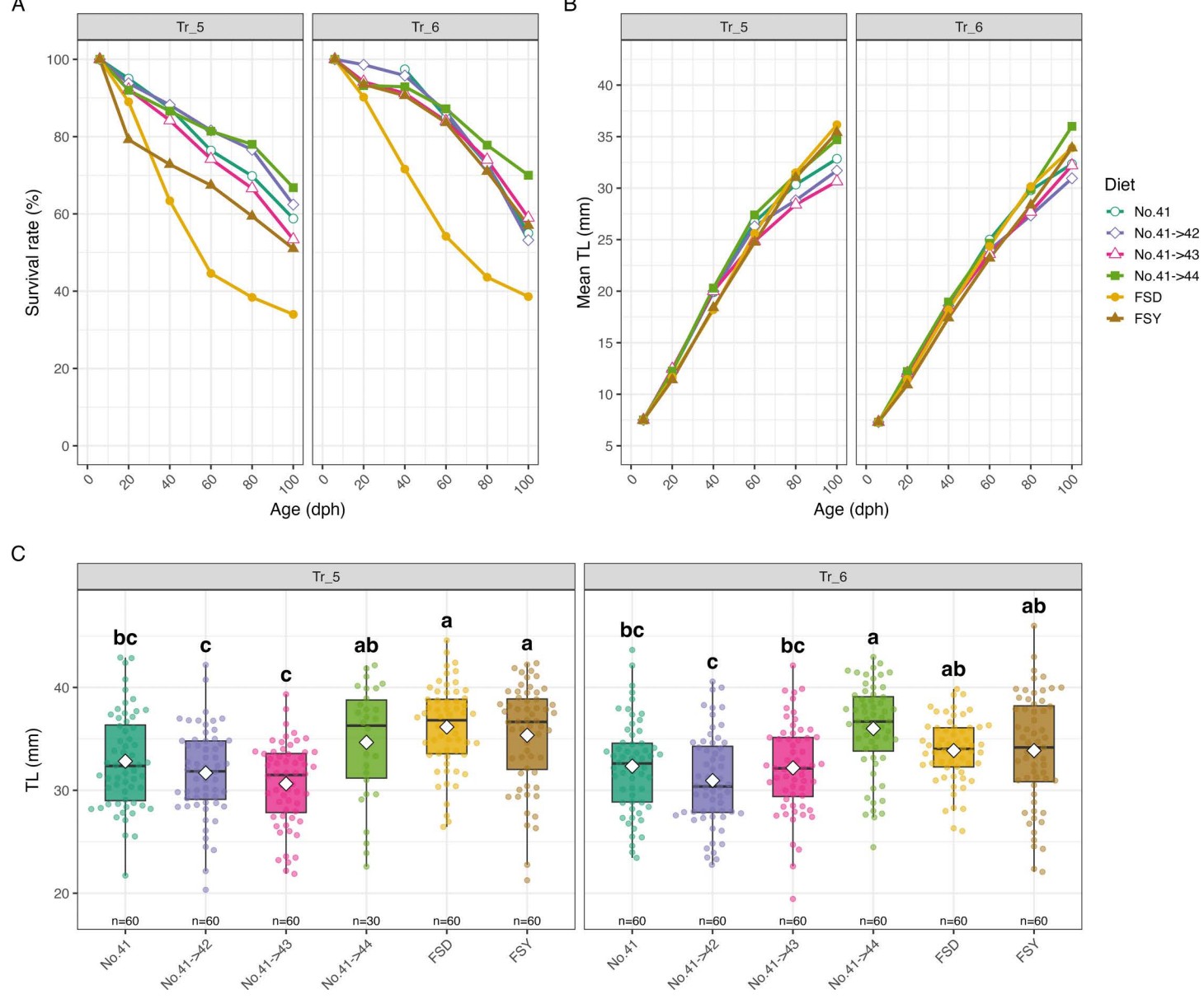

**Fig 4. Comparison of survival rate and growth among diet groups in validation trials (Tr_5-6). (A)** Survival rate and **(B)** mean TL from 6 to 100 dph, shown as line plots by diet group in each trial. Data points represent the mean values from replicate tanks for each diet at each sampling time point. **(C)** Box plots of TL at 100 dph for each diet group in Tr_5 and Tr_6. Boxes indicate the interquartile range, the horizontal line represents the median, and white diamonds indicate the mean. Different lowercase letters above the boxes denote significant differences among diets within each trial (Tukey-Kramer test, $p < 0.05$). Sample sizes (n) for each group are indicated above the x-axis. **Note:** In Tr_5, due to a tank outflow accident in one replicate of the No.41→44 group at 76 dph, only one tank was available for data collection after 80 dph in this group.

To further verify reproducibility and robustness, we refined the candidate diets based on the Tr_5–6 results, narrowing the final validation trials (Tr_7–8) to Nos. 41 and 44. In these trials, we directly compared diet No. 41 alone, diet No. 44 alone, and the No. 41→44 combination, as well as the reference diets (including a sequential switch from FSD to FSY [FSD→FSY]), to further evaluate the effects of switching diets at 40 dph. In Tr_5–6, the No. 41→44 group had achieved

higher survival rates than the reference diets, which resulted in relatively higher stocking densities during the late rearing phase, raising concerns about potential density-related growth bias. To address this, larval densities were equalized at 41 dph across treatments.

In Tr_7, No. 41→44 again achieved the highest mean survival rate (74.9%), followed by No. 44 (72.4%) and the reference diets (FSY: 63.8%; FSD: 43.4%). In Tr_8, diet No. 44 yielded the highest survival rate (58.0%), followed by No. 41→44 (52.9%) and the reference diets (FSY: 36.8%; FSD: 18.0%) (Fig 5A). For mean TL at 100 dph, the optimized and reference diets showed similar results in Tr_7 (No. 41→44: 37.9±0.53 mm; FSY: 38.4±0.45 mm; FSD: 34.5±0.44 mm), with Tukey-Kramer tests indicating that No. 41, No. 44 and No. 41→44 were in the top statistical groups, but not always distinct from the reference diets. In Tr_8, the No. 41→44 diet also achieved a high TL (40.1±0.51 mm), comparable to or exceeding the reference diets (FSY: 38.4±0.72 mm; FSD: 35.9±0.65 mm) (Fig 5B, C).

Overall, the BO-recommended diets, particularly No. 41, No. 44, and No. 41→44, consistently achieved high survival and growth performance in the final validation trials, confirming the robustness and generalizability of the optimized formulations across independent experiments under strictly controlled conditions.

## Key Drivers of Larval Performance: RF model and SHAP Analysis

To clarify which nutritional factors most strongly influenced larval survival and growth, we used RF regression models with SHAP values and analyzed the estimated proximate composition of the experimental diets across all eight feeding trials (Tr_1–8). Separate models were constructed for each developmental period (6–40 and 41–80 dph) and for each objective variable (survival rate and mean TL), with explanatory variables consisting of crude protein, crude lipid, crude ash, crude carbohydrate, and nucleic acids (expressed as % of dry matter). Scatter plots illustrating the relationships between each nutrient component and standardized survival rate or mean TL for all diets and periods are shown in S2 Fig. The predictive performance of the RF models is shown in S3 Fig, which displays scatter plots of predicted versus observed standardized z-scores for the survival rate and mean TL in both developmental periods, based on 10-fold cross-validation.

As shown in Table 4, the relative importance of these components in contributing to model predictions varied depending on the period and trait. For survival predictions during the early rearing period (6–40 dph), nucleic acids were the most influential components (24.0%), followed by crude ash (20.2%), crude carbohydrates (20.2%), and crude lipids (19.7%). In contrast, for growth (mean TL) predictions during the same period, crude ash was the most important factor (28.4%), followed closely by nucleic acids (28.3%), with crude carbohydrates (16.8%) and crude proteins (16.5%) also contributing substantially. In the later rearing period (41–80 dph), the key drivers for survival rate predictions shifted, with crude ash (38.2%) and nucleic acids (36.2%) being dominant, whereas crude proteins (11.0%), crude carbohydrates (8.4%), and especially crude lipids (6.2%) played smaller roles. For mean TL predictions in this later period, crude lipids were the primary factor (27.7%), followed by crude protein (25.2%), nucleic acids (20.4%), carbohydrates (14.6%) and crude ash (12.2%) with lesser influence.

The SHAP dependence plots (Fig 6) further visualize these relationships; for example, high protein content (65–75% DM) was associated with improved survival during the early phase, whereas moderate protein content (55–60% DM) was optimal in the later phase. Regarding lipid content, although growth contributions peaked around 25% DM in the later phase, the model suggested approximately 15–20% DM as the balanced effective range to ensure survival stability. Ash and nucleic acid contents also showed positive effects, especially in the later stages, with their influences on survival predictions plateauing at roughly 8% and 4%, respectively. Notably, the relative importance of each nutrient component changed substantially with larval age and target traits, highlighting the stage-specific and multifactorial nature of dietary optimization in Japanese eel larval rearing.

Overall, these analyses quantitatively demonstrate that the enhanced survival and growth observed with BO-recommended diets can be attributed to the precise balancing of multiple nutritional factors and provide a mechanistic foundation for the formulating optimized diets tailored to specific developmental stages.

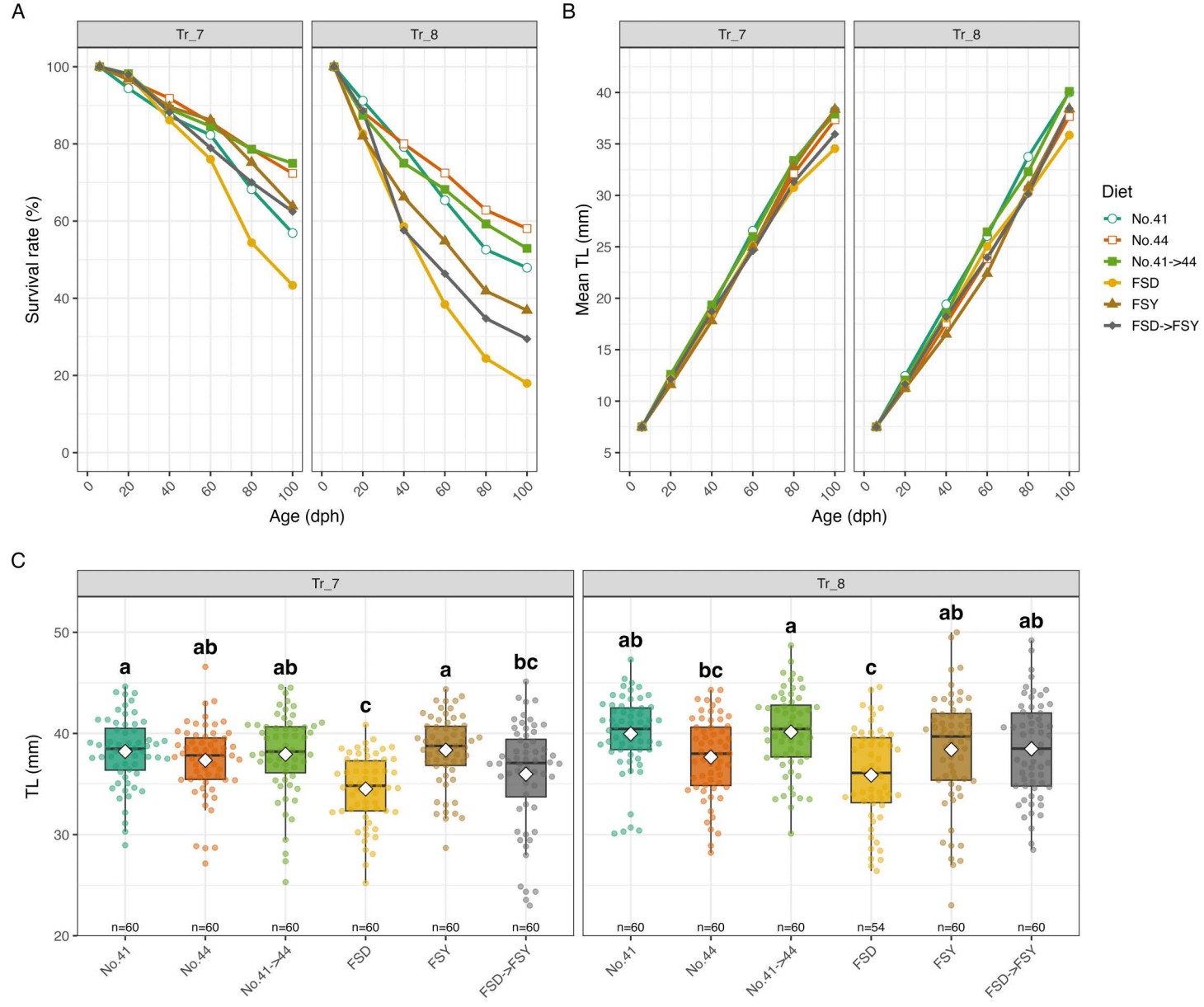

**Fig 5. Comparison of survival rate and growth among diet groups in final validation trials (Tr_7-8). (A)** Survival rate and **(B)** mean TL from 6 to 100 dph, shown as line plots by diet group in each trial. Data points represent the mean values from replicate tanks (n=2) for each diet at each sampling time point. **(C)** Box plots of TL at 100 dph for each diet group in Tr_7 and Tr_8. Boxes indicate the interquartile range, the horizontal line represents the median, and white diamonds indicate the mean. Different lowercase letters above the boxes denote significant differences among diets within each trial (Tukey-Kramer test, $p < 0.05$). Sample sizes (n) for each group are indicated above the x-axis. **Note:** In Tr_7 and Tr_8, the stocking density was adjusted at 41 dph. Therefore, after 41 dph, the survival rate was calculated by multiplying the cumulative survival rate up to 40 dph by the interval survival rate after 41 dph, using the number of larvae stocked at 41 dph as the new denominator.

## Discussion

This study provides robust evidence that data-driven adaptive optimization using BO can significantly improve larval feed formulations for the Japanese eel. The development of innovative larval diets remains a critical challenge in artificial seedling production. Traditionally, eel larval diet development has relied on stepwise adjustments of basic formulations

**Table 4. Variable importance of proximate composition components in the random forest model fitted to data from Tr_1-8.**

| Period | Objective variables | Crude protein | Crude lipid | Crude ash | Crude carbohydrate | Nucleic acids |
|---|---|---|---|---|---|---|
| 6-40 dph | Survival rate | 15.9% | 19.7% | 20.2% | 20.2% | 24.0% |
| | mean TL | 16.5% | 10.0% | 28.4% | 16.8% | 28.3% |
| 41-80 dph | Survival rate | 11.0% | 6.2% | 38.2% | 8.4% | 36.2% |
| | mean TL | 25.2% | 27.7% | 12.2% | 14.6% | 20.4% |

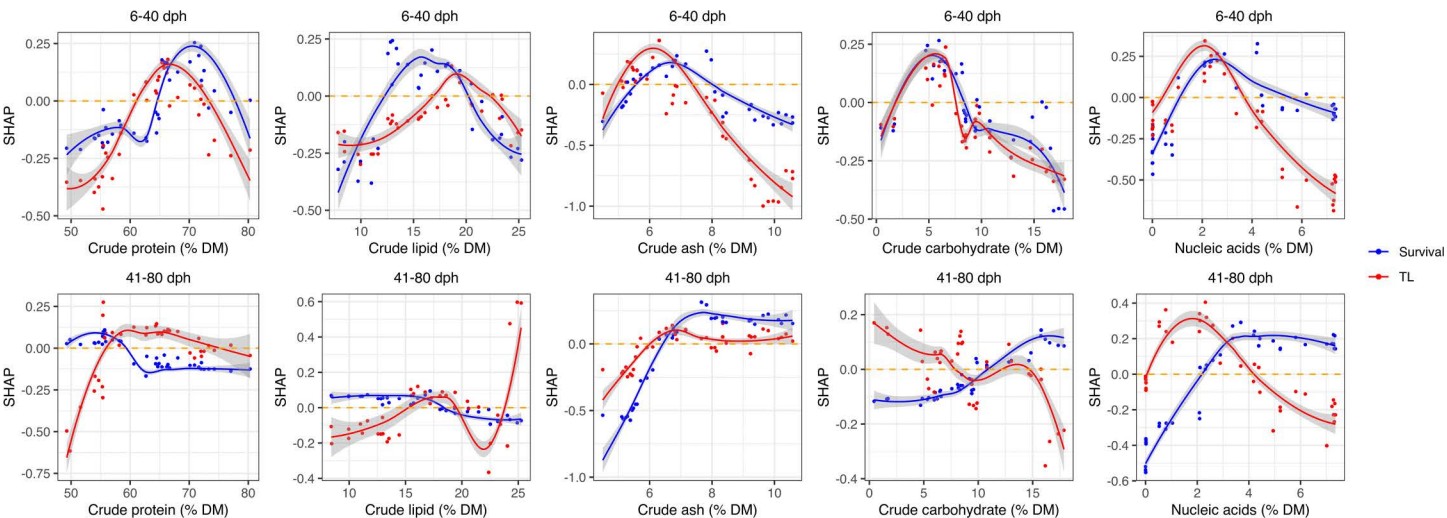

**Fig 6. SHAP dependence plots illustrating the effects of dietary proximate composition on standardized survival rate (blue) and mean TL (red) of Japanese eel larvae in the early (6–40 dph, upper row) and late (41–80 dph, lower row) rearing periods.** Each panel shows the relationship between the value of a given dietary component (x-axis) and its SHAP value (y-axis) as predicted by RF regression models trained for each period and trait, using data from all eight feeding trials (Tr_1–8). Points represent individual diets; loess curves indicate local trends, with shaded areas denoting 95% confidence intervals.

derived from shark egg yolk-based diets or their alternatives, mainly through simple heuristic methods, such as iterative hypothesis testing and trial and error [5,6,20–23]. Although such approaches offer advantages in terms of simplicity, practicality, and ease of implementation, they have inherent limitations in efficiently exploring complex, high-dimensional formulation spaces. In contrast, the introduction of BO in this study enabled a systematic and efficient search across a multidimensional diet composition space, allowing for the simultaneous optimization of multiple objectives, namely, survival and growth. The use of GPR models enabled the explicit quantification of uncertainty and facilitated exploration not only of known high-performing formulations but also of previously untested regions of the formulation space. Although mixture-design-based frameworks [10] have previously been used to optimize macronutrient composition in finfish diets, those approaches relied on static experimental designs and limited iteration. Our BO framework extends this concept into a sequential, probabilistic domain that dynamically updates the experimental design based on prior results, enabling efficient exploration of high-dimensional formulation spaces. Although metaheuristic techniques have recently attracted attention in animal diet optimization [11], to our knowledge, this is the first practical report of BO being applied to larval feed optimization for the Japanese eel.

Ensuring experimental reproducibility presents a major challenge in larval rearing trials of Japanese eels, largely due to pronounced variability in survival among rearing batches and tanks. Under ideal conditions, rigorous replication across tanks and independent batches would be required to fully account for such variability. However, practical constraints such

as limited tank availability, long experimental durations extending to metamorphosis, and high operational costs render exhaustive replication infeasible during early-stage feed screening. To address this challenge, we adopted a stepwise and adaptive experimental design that explicitly prioritized exploratory breadth in the initial phases while reserving rigorous replication for final validation. During the baseline data collection (Tr_1–2) and BO exploration phases (Tr_3–4), a larger number of candidate formulations were screened using single-tank trials, with batch-level normalization achieved through the inclusion of standard reference diets. Rather than relying solely on static replication, robustness was enhanced through the adaptive nature of BO, whereby promising formulations were repeatedly sampled across independent trials as the search space narrowed. This cumulative evaluation across multiple rearing batches reduced the likelihood that optimization outcomes were driven by single-trial outliers or lot-specific biases. Importantly, the reproducibility and performance of the selected formulations were subsequently confirmed in the validation phase (Tr_5–8) using replicated trials, supporting the overall validity of this design strategy.

In the initial baseline collection phase (Tr_1–2), no diet formulation clearly outperformed the reference diets; however, in Tr_3, formulation No. 31 resulted in notable improvements in both survival and growth. Therefore, in Tr_4, we re-evaluated No. 31 along with similar BO-recommended candidates, confirming further enhancements in both metrics. Notably, diet No. 31 did not include soy peptide or CPSP, but had a higher proportion of casein sodium compared to the reference diets, a formulation that had already demonstrated good performance in preliminary trials. This study adopts a human-in-the-loop Bayesian optimization framework that integrates empirical knowledge and expert judgment, rather than relying solely on automated black-box optimization. Incorporating expert insight allows for a more efficient narrowing of the search space and selection of promising candidates, thereby reducing the number of required trials, a strategy that has proven effective in other fields, such as pharmaceutical and material development [24]. For high-cost, long-term larval rearing experiments, such as those for the Japanese eel, combining data-driven methods with domain expertise is key to achieving practical and efficient optimization, while also facilitating real-world implementation by researchers.

In multiple validation trials (Tr_5–8), the BO-optimized diets, particularly Nos. 41 and 44, and the sequential regimen (No. 41→44), consistently achieved higher survival rates and growth rates than the conventional and reference diets. By introducing a density adjustment at 41 days post-hatch, we also avoided the risk of underestimating growth in the high-survival groups due to density bias, thereby demonstrating the performance of BO-optimized diets. Jinbo et al. (2025) reported that rearing with an FSD resulted in spinal deformities in approximately 30% of glass eels, whereas supplementation with a yeast extract rich in nucleic acids (FSY) reduced this rate to approximately 3% [6]. Notably, Nos. 41 and 44 also contained yeast extract and were therefore expected to exhibit a similar suppressive effect on skeletal deformities. Even after rearing the larvae through metamorphosis, the frequency of spinal deformities observed with No. 41 and No. 41→44 was comparable to that with FSY (S4 Table), indicating that there were no negative impacts on the quality of the glass eels.

The RF model and SHAP analysis provide mechanistic insights into the nutritional drivers underlying the observed improvements. Importantly, the influence of each dietary component on model predictions varied depending on developmental stage and objective variable. During the early period, nucleic acid, ash, carbohydrate, and lipid contents had similarly influences on survival prediction, whereas nucleic acids and ash contributed most to growth prediction. In the later period, nucleic acids and ash became the dominant factors for survival prediction, whereas lipids and proteins were the most relevant to growth. Notably, ash content contributed substantially to survival prediction during the early period and to growth predictions during the later period. In these diets, ash is largely derived from yeast extracts. Thus, optimization of the yeast extract content was a major factor underlying the improvements in larval performance according to the model. These findings highlight that the optimal range for each nutrient shifts with larval stage and target trait, emphasizing the necessity of period-specific and multifactorial diet design. Although not quantitatively analyzed here, the physical properties of the diets, such as viscosity and particle size, were carefully controlled. These properties are known to affect feed intake and should be considered in future optimization efforts. The present model is limited to a fixed set of ingredients

and does not incorporate cost or ingredient availability constraints. Future research should pursue multi-objective optimization, incorporating economic and sustainability factors, and may benefit from integrating BO with other machine learning approaches to further enhance prediction accuracy and model interpretability. Furthermore, employing multi-omics approaches (e.g., transcriptomics and metabolomics) alongside physiological assessments would provide valuable insights into the metabolic pathways and biological mechanisms driven by the optimized diets.

Improved growth performance with increased dietary protein levels is common in most fish; however, excessive protein inclusion in the diet may lead to reduced growth. The high energy demand for catabolism, rather than protein deposition, can result in reduced growth rates when diets are excessively high in protein [25]. Growth reduction associated with high protein content may result from the accumulation of toxic nitrogen compounds that impair development [26,27]. The weight gain and total length of larval rockfish (*Sebastes schlegeli*) increased with dietary protein levels up to 54%, but decreased with further increases in crude protein content [28]. Larval sea bass (*Dicentrarchus labrax*) fed a diet containing 60% protein showed slightly lower growth than those fed a 50% protein diet [29]. Late larval or early juvenile Atlantic cod (*Gadus morhua*) fed diets containing more than 60% protein also showed reduced growth [30]. These results indicate that the optimal dietary protein level for fish larvae is approximately 50–60%. In the present study, the growth of eel larvae during the early stage was optimal at a dietary protein content of approximately 65%, which is higher than the optimal levels reported for other species. Eel larvae in an early stage may possess a greater capacity to utilize protein than larvae of other fish species and may require higher protein levels for growth. In contrast, the growth of larvae during the later period improved with dietary protein content up to 55%, but did not increase further with higher protein levels. For other nutrients, such as with ash, the relationship between dietary content and growth became less pronounced in the later stages compared with the early stages. In Atlantic cod, growth during the late larval stage is influenced by dietary nutrient balance (protein, lipid, and carbohydrate), whereas juvenile growth is not significantly affected by diet composition [30]. Nutrient requirements in eels may change with growth during the larval stage. Further research is needed to clarify the stage-specific nutritional requirements of eel larvae and to develop diets tailored to each growth stage.

In summary, our data-driven, human-in-the-loop Bayesian optimization framework successfully developed larval diets that significantly improved survival and growth of Japanese eel larvae without shark-derived ingredients. Model-based analyses suggested the key role of yeast extract-derived nutrients and macronutrient balances, with nutritional requirements varying by developmental stage. This adaptive framework can accelerate feed innovation in eel aquaculture and offers broad applicability to larval feed development in other species.

## Conclusion

This study demonstrated that Bayesian optimization, combined with mechanistic modeling and stage-specific analysis, can efficiently improve larval feed formulations for the Japanese eel (*Anguilla japonica*). The resulting optimized diets, characterized by a carefully balanced composition of proteins, lipids, and nucleic acids from yeast extract, consistently enhanced both survival and growth in replicated validation trials. Our findings further highlight the critical importance of multifactorial and stage-adapted dietary optimization and offer a practical framework for advancing the sustainability and productivity of artificial seed production in eel aquaculture, with broad applicability to other aquaculture species.

## Supporting information

**S1 Fig. Scatter plots of predicted versus observed values from 10-fold cross-validation for four GPR models fitted to data from Tr_1–4.** Each panel shows the relationship between predicted and observed standardized z-scores for survival rate or mean total length (TL) during two developmental periods (6–40 dph and 41–80 dph). Predictions were obtained using 10-fold cross-validation.
(PDF)

**S2 Fig. Scatter plots of standardized survival rate and mean total length (TL) against estimated proximate composition (% of dry matter) of diet formulations in Tr_1–8.** Each panel shows the relationship between the percentage of crude protein, crude lipid, crude ash, crude carbohydrate, or nucleic acids in the diet (x-axis) and the standardized objective variables (y-axis; survival rate or mean TL) for each diet. Results are shown separately for the two developmental periods (6–40 dph and 41–80 dph).
(PDF)

**S3 Fig. Scatter plots of predicted versus observed values from 10-fold cross-validation for four RF models fitted to data from Tr_1–8 with explanatory variables consisting of crude protein, crude lipid, crude ash, crude carbohydrate, and nucleic acids (% of dry matter).** Each panel shows the relationship between predicted and observed standardized z-scores for survival rate or mean total length (TL) during two developmental periods (6–40 dph and 41–80 dph). Predictions were obtained using 10-fold cross-validation.
(PDF)

**S1 Table. Ingredient boundaries (lower and upper limits, %) used for D-optimal design and Bayesian optimization in each trial.**
(XLSX)

**S2 Table. Proximate composition (%) for each ingredient.**
(XLSX)

**S3 Table. Estimated proximate composition (% of Dry matter) for each experimental diets.**
(XLSX)

**S4 Table. Spinal deformity frequency in metamorphosed glass eels per diet after extended rearing following Tr_5 and Tr_6.**
(XLSX)

**S1 File. Python scripts and datasets used for Bayesian optimization and random forest analyses.**
(ZIP)

## Acknowledgments

We would like to thank the staff of the Shibushi Field Station of the Fisheries Technology Institute, Japan Fisheries Research and Education Agency, for their invaluable assistance in rearing the fish used in this study.

## Author contributions

**Conceptualization:** Kazuharu Nomura, Tadao Jinbo, Hirofumi Furuita, Masato Higuchi.

**Data curation:** Kazuharu Nomura, Tadao Jinbo, Hirofumi Furuita, Masato Higuchi, Taiga Asakura.

**Formal analysis:** Kazuharu Nomura, Tadao Jinbo.

**Investigation:** Kazuharu Nomura, Tadao Jinbo, Hirofumi Furuita, Masato Higuchi, Hiroshi Suzuki, Takashi Yatabe.

**Methodology:** Kazuharu Nomura, Tadao Jinbo, Hirofumi Furuita, Masato Higuchi.

**Project administration:** Kazuharu Nomura, Tadao Jinbo.

**Resources:** Tadao Jinbo.

**Validation:** Kazuharu Nomura, Tadao Jinbo, Hirofumi Furuita, Taiga Asakura.

**Visualization:** Kazuharu Nomura.

**Writing – original draft:** Kazuharu Nomura.

**Writing – review & editing:** Kazuharu Nomura, Tadao Jinbo, Hirofumi Furuita, Masato Higuchi, Taiga Asakura, Hiroshi Suzuki, Takashi Yatabe, Miyuki Mekuchi, Takeshi Hano, Takashi Ishikawa, Youhei Fukui, Nobuto Kaneko.

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
