## [Decision Letter · Decision Letter 0]

23 Dec 2025

*Anguilla japonica*

Dear Dr. Nomura,

Thank you for submitting your manuscript to PLOS ONE. After careful consideration, we feel that it has merit but does not fully meet PLOS ONE’s publication criteria as it currently stands. Therefore, we invite you to submit a revised version of the manuscript that addresses the points raised during the review process.

1. This manuscript not technically sound, and the data cannot support the conclusions. PLOS ONE is designed to communicate primary scientific research, and welcome submissions in any applied discipline that will contribute to the base of scientific knowledge. But this manuscript not adhere to the criteria for scientific research article that results show not sufficient to support the conclusion.

2. The revised manuscript needs to address each of the comments of the reviewers.

We look forward to receiving your revised manuscript.

Kind regards,

Tzong-Yueh Chen, Ph.D.

Academic Editor

PLOS One

Additional Editor Comments (if provided):

Reviewers' comments:

Reviewer's Responses to Questions

**Comments to the Author**

1. Is the manuscript technically sound, and do the data support the conclusions?

Reviewer #1: Partly

Reviewer #2: Yes

2. Has the statistical analysis been performed appropriately and rigorously?

Reviewer #1: Yes

Reviewer #2: Yes

3. Have the authors made all data underlying the findings in their manuscript fully available?

Reviewer #1: Yes

Reviewer #2: Yes

4. Is the manuscript presented in an intelligible fashion and written in standard English?

Reviewer #1: Yes

Reviewer #2: Yes

Reviewer #1: The article entitled “Data-Driven Optimization of Diet Formulation to Enhance Survival and Growth in Japanese Eel (Anguilla japonica) Larvae” employs Bayesian optimization together with Gaussian process regression to establish a data-driven framework for larval diet refinement. This approach consistently improved the performance of shark egg yolk–free formulations across multiple trials, elevating survival and growth. Through mechanistic modeling, the authors further identified that stage-specific balancing of yeast extract–derived nutrients and macronutrient components is a key determinant of larval performance. This study represents the first application of BO in diet development for Japanese eel, effectively reducing experimental effort while demonstrating high reproducibility and cross-species applicability. Overall, the work offers clear innovation and strong publication potential. However, several minor comments still need to be addressed before the manuscript can be accepted for publication.

Minor comments:

1. The content in Lines 171–211 describing the D-optimal design and BO procedure should be integrated into the four-stage section in Lines 134–153, rather than presented separately.

2. Lines 177-178: What criteria were used for manually adjusting the feed formulations, and were any reference sources consulted?

3. Line 180: Why were the feed formulations numbered as Nos. 13–22 instead of Nos. 9–18?

4. In S1 Table, were the upper and lower boundaries for each ingredient reversed?

5. Lines262-264: The procedure for calculating the z score has already been described in the Materials and Methods and does not need to be repeated here. In addition, why was the overall period TL z score not calculated for diet No. 38? Since z scores were calculated for both the early and late periods, it should theoretically be possible to derive a z score for the overall period as well.

6. Lines307-308: What was the rationale for selecting the feed formulations Nos. 42–44? These three formulations appear different but still share a certain degree of similarity. Why were two or three additional formulations with greater differences from these three not selected for testing?

7. Line 337、366 : below should be revised to above.

8. Line 354、371: Based on Fig. 5C, in addition to No. 41→44 and No. 44, it would be more reasonable to include No. 41 as well.

9. Lines367-369: However, why is there such a large difference in the mean survival rates between Tr_7 and Tr_8? Why is the survival rate in Tr_8 substantially lower?

10. Line 408: Based on Figure 6, I would argue that the optimal lipid content for growth was approximately 15–20%.

11. Line 451: This clearly indicates that larval survival rates varied significantly among different rearing batches, and therefore at least two independent replicate experiments are required to reliably assess and confirm reproducibility. However, why were two independent replicate experiments not conducted for Tr_1–Tr_4? Could this lead to potential bias in the selection of feed formulations during the baseline data collection and BO exploration phases?

12. Lines479-489: This paragraph should include a discussion of the results of feed group No. 41 as well.

13. Line 525: Should {such as with “proteins” }be changed to {such as with “ash” }?

14. Line 543: suggesting “These findings” to be changed to “Our findings further”

15. Lines 568-571: The cited article provides insufficient information.

Reviewer #2: The authors implemented a data-driven, human-in-the-loop optimization framework based on Bayesian optimization and Gaussian process regression to systematically refine shark egg yolk-free diets to overcome the limited efficacy of conventional heuristic formulation methods. The present study was worth to be accepted and provided an insight for eel cultivation. The further estimations of the resulting optimized diets on growth or physiological responses were suggested to be conducted to prove the feasibility of this formula.

**Do you want your identity to be public for this peer review?** For information about this choice, including consent withdrawal, please see our Privacy Policy

Reviewer #1: No

Reviewer #2: No

---

## [Author Response · Author response to Decision Letter 1]

13 Jan 2026

Manuscript ID: PONE-D-25-57759

Title: Data-driven optimization of diet formulation to enhance survival and growth in Japanese eel (Anguilla japonica) larvae

Dear Editor and Reviewers,

We sincerely thank you for the time and effort devoted to the review of our manuscript. We greatly appreciate the constructive comments and suggestions, which have helped us improve both the quality and clarity of the work.

We have carefully revised the manuscript in accordance with all comments. Point-by-point responses are provided below, and all changes are highlighted in the “Track Changes” version of the revised manuscript.

We hope that these revisions adequately address the concerns raised and that the manuscript is now suitable for publication in PLOS ONE.

Sincerely,

Kazuharu Nomura

(On behalf of all co-authors)

Response to the Editor

Comment 1

Response:

We have carefully reviewed the manuscript and ensured compliance with all PLOS ONE style requirements. Specifically, heading styles, font sizes, and file naming conventions for figures and supporting information have been revised accordingly.

Comment 2

Please note that, though access restrictions are acceptable now, your entire data will need to be made freely accessible if your manuscript is accepted for publication.

Response:

We appreciate the Editor’s recommendation to enhance transparency and reproducibility. In response, we have made all relevant data freely accessible by providing the Python scripts used for analysis together with the corresponding datasets as a Supplementary File (S1 File). The “Data Availability Statement” and “Materials and Methods” sections have been updated accordingly.

Response to Reviewer #1

Comment 1

The content in Lines 171–211 describing the D-optimal design and BO procedure should be integrated into the four-stage section in Lines 134–153, rather than presented separately.

Response:

We agree that integrating these methodological details improves clarity and logical flow. The “Experimental design and adaptive optimization process” section has been revised so that the D-optimal design and Bayesian optimization procedures are described directly within Phases I–IV (Revised manuscript, Lines 134–189).

Comment 2

Lines 177-178: What criteria were used for manually adjusting the feed formulations, and were any reference sources consulted?

Response:

Manual adjustments were guided by (i) physicochemical feasibility (e.g., slurry consistency suitable for larval ingestion and stability in water) and (ii) biological safety (avoiding extreme nutrient imbalances), based on expert knowledge and prior eel larval rearing studies. These criteria are now explicitly stated in the “Materials and Methods” section (Revised manuscript, Lines 139–143).

Comment 3

Line 180: Why were the feed formulations numbered as Nos. 13–22 instead of Nos. 9–18?

Response:

The numbering reflects internal laboratory IDs used for traceability. Diets Nos. 9–12 were tested in preliminary trials but excluded from the present dataset due to unrelated technical issues. This explanation has been added to the “Materials and Methods” section (Revised manuscript, Lines 146–147).

Comment 4

In S1 Table, were the upper and lower boundaries for each ingredient reversed?

Response:

We thank the reviewer for identifying this error. The upper and lower boundaries were indeed reversed, and S1 Table has been corrected accordingly.

Comment 5

Lines262-264: The procedure for calculating the z score has already been described in the Materials and Methods and does not need to be repeated here. In addition, why was the overall period TL z score not calculated for diet No. 38? Since z scores were calculated for both the early and late periods, it should theoretically be possible to derive a z score for the overall period as well. Response:

We agree that the description of the z-score calculation was redundant and have removed the repeated sentences. The overall TL z-score for Diet No. 38 (6–80 dph) was calculated; however, because the value was effectively zero, the bar is not visually discernible in Fig 1. This reflects performance very close to the mean rather than missing data.

Comment 6

Lines307-308: What was the rationale for selecting the feed formulations Nos. 42–44? These three formulations appear different but still share a certain degree of similarity. Why were two or three additional formulations with greater differences from these three not selected for testing?

Response:

The selection of three candidates (Nos. 42–44) was primarily constrained by experimental resources. To ensure robustness using duplicate tanks in the validation phase, only three new candidates could be tested after allocating reference and baseline groups. Under these constraints, we prioritized exploitation of the predicted optimal region rather than broader exploration. Diets Nos. 42–44 were selected to represent this region by balancing high-importance ingredients (yeast extract, casein Na, skim milk, and CPSP). This rationale is now clarified in the “Results” section (Revised manuscript, Lines 300–303).

Comment 7:

Line 337、366 : below should be revised to above.

Response:

Corrected as suggested (Revised manuscript, Lines 331 and 360).

Comment 8

Line 354、371: Based on Fig. 5C, in addition to No. 41→44 and No. 44, it would be more reasonable to include No. 41 as well.

Response:

We agree. Diet No. 41 has been added to the relevant description in the “Results” section (Revised manuscript, Lines 348 and 365).

Comment 9

Lines367-369: However, why is there such a large difference in the mean survival rates between Tr_7 and Tr_8? Why is the survival rate in Tr_8 substantially lower?

Response:

This difference is attributable to lot effects (inter-batch variability), including differences in egg quality, parental genetics, and environmental conditions. This point has been expanded in the “Discussion” section (Revised manuscript, Lines 444–445).

Comment 10

Line 408: Based on Figure 6, I would argue that the optimal lipid content for growth was approximately 15–20%.

Response:

We agree. Re-examination of Fig. 6 confirmed consistently positive SHAP contributions to both growth and survival in the 15–20% DM range. Although higher lipid levels (25%) showed peaks for growth in later phases, they negatively affected survival predictions. The text has been revised accordingly (Revised manuscript, Lines 401–403).

Comment 11

Line 451: This clearly indicates that larval survival rates varied significantly among different rearing batches, and therefore at least two independent replicate experiments are required to reliably assess and confirm reproducibility. However, why were two independent replicate experiments not conducted for Tr_1–Tr_4? Could this lead to potential bias in the selection of feed formulations during the baseline data collection and BO exploration phases?

Response:

We appreciate this important point. While independent replicates are ideal given the pronounced lot-to-lot variability, we prioritized exploration speed over evaluation precision during the screening phase to sample a wider formulation space. Replication at this stage would have substantially reduced the number of formulations tested. To mitigate potential bias, we (i) applied normalization using in-trial reference diets (z-scores) to correct baseline shifts and (ii) leveraged the iterative nature of Bayesian optimization, whereby promising regions are repeatedly sampled across independent batches. This cumulative re-evaluation increases robustness against lot-specific outliers. The rationale is now more clearly articulated in the “Discussion” section (Revised manuscript, Lines 444–460).

Comment 12

Lines479-489: This paragraph should include a discussion of the results of feed group No. 41 as well.

Response:

We agree and have revised the paragraph to explicitly discuss Diet No. 41 and the sequential regimen (No. 41→44) (Revised manuscript, Lines 475–476).

Comment 13

Line 525: Should {such as with “proteins” }be changed to {such as with “ash” }?

Response:

Corrected as suggested (Revised manuscript, Line 524).

Comment 14

Line 543: suggesting “These findings” to be changed to “Our findings further”.

Response:

Revised accordingly (Revised manuscript, Line 542).

Comment 15

Lines 568-571: The cited article provides insufficient information.

Response:

We apologize for the incomplete citation and have provided full bibliographic details in the revised manuscript (Revised manuscript, Line 570).

Response to Reviewer #2

Comment 1

The present study was worth to be accepted and provided an insight for eel cultivation. The further estimations of the resulting optimized diets on growth or physiological responses were suggested to be conducted to prove the feasibility of this formula.

Response:

We sincerely thank the reviewer for the positive assessment. We fully agree that evaluating physiological responses is essential to confirm the biological feasibility of the optimized diets. We are currently planning follow-up studies incorporating multi-omics approaches (e.g., transcriptomics and metabolomics) to elucidate underlying mechanisms. While the present study focused on establishing the optimization framework and phenotypic outcomes, we have added a statement in the “Discussion” section highlighting the importance of such integrated analyses in future work (Revised manuscript, Lines 505–507).

---

## [Decision Letter · Decision Letter 1]

2 Feb 2026

Data-Driven Optimization of Diet Formulation to Enhance Survival and Growth in Japanese Eel (*Anguilla japonica* ) Larvae

PONE-D-25-57759R1

Dear Dr. Nomura,

We’re pleased to inform you that your manuscript has been judged scientifically suitable for publication and will be formally accepted for publication once it meets all outstanding technical requirements.

Kind regards,

Tzong-Yueh Chen, Ph.D.

Academic Editor

PLOS One

Additional Editor Comments (optional):

Reviewers' comments:

Reviewer's Responses to Questions

**Comments to the Author**

Reviewer #2: All comments have been addressed

2. Is the manuscript technically sound, and do the data support the conclusions?

Reviewer #2: Yes

3. Has the statistical analysis been performed appropriately and rigorously?

Reviewer #2: Yes

4. Have the authors made all data underlying the findings in their manuscript fully available?

Reviewer #2: Yes

5. Is the manuscript presented in an intelligible fashion and written in standard English?

Reviewer #2: Yes

Reviewer #2: Although the estimations of the resulting optimized diets on growth or physiological responses were unavailable in the present study to prove the feasibility of this formula, yet the authors are currently planning follow-up studies incorporating multiomics approaches. This is acceptable.

**Do you want your identity to be public for this peer review?** For information about this choice, including consent withdrawal, please see our Privacy Policy

Reviewer #2: No

---

## [Editor Report · Acceptance letter]

PONE-D-25-57759R1

PLOS One

Dear Dr. Nomura,

I'm pleased to inform you that your manuscript has been deemed suitable for publication in PLOS One. Congratulations! Your manuscript is now being handed over to our production team.

Kind regards,

on behalf of

Prof. Tzong-Yueh Chen

Academic Editor

PLOS One